

# Field Evaluation of Low-cost Electrochemical Air Quality Gas Sensors at Extreme Temperature and Relative Humidity Conditions

Roubina Papaconstantinou[1], Marios Demosthenous[1], Spyros Bezantakos[1], Neoclis Hadjigeorgiou[1], Marinos Costi[1], Melina Stylianou[2], Elli Symeou[2], Chrysanthos Savvides[3], George Biskos[1,4]

[1]Climate and Atmosphere Research Centre, The Cyprus Institute, Nicosia 2121, Cyprus
[2]Medisell Co Ltd, Nicosia 2033, Cyprus
[3]Ministry Labour Welfare and Social Insurance, Department of Labour Inspection, Nicosia 1463, Cyprus
[4]Faculty of Civil Engineering and Geosciences, Delft University of Technology, Delft 2628CN, The Netherlands

*Correspondence to*: George Biskos (g.biskos@cyi.ac.cy)

**Abstract.** Modern electrochemical gas sensors hold great potential for improving practices in Air Quality (AQ) monitoring as their low cost, ease of operation and compact design can enable dense observational networks and mobile measurements. Despite that, however, numerous studies have shown that the performance of these sensors depends on a number of factors (e.g., environmental conditions, sensor quality, maintenance and calibration, etc.), thereby adding significant uncertainties in the reported measurements and large discrepancies from those recorded by reference-grade instruments. In this work we

investigate the performance of electrochemical sensors, provided by two manufacturers (namely Alphasense and Winsen), for measuring the concentrations of $CO$, $NO_2$, $O_3$ and $SO_2$. To achieve that we carried out collocated yearlong measurements with reference-grade instruments at a traffic AQ monitoring station in Nicosia, Cyprus, where temperatures range from almost 0 °C in the winter, to almost 45 °C in the summer. The CO sensors exhibit the best performance among all we tested, having minimal mean relative error (MRE) compared to reference instruments (ca. 5%), although a significant difference in their response was

observed before and after the summer period. At the other end of the spectrum, the $SO_2$ sensors reported concentration values that were at least one order of magnitude higher than the respective reference measurements (with MREs being more than 1000% for Alphasense and almost 400% for Winsen throughout the entire measuring period), which can be justified by the fact that the concentrations of $SO_2$ at our measuring site were below their Limit of Detection. In general, variabilities in the environmental conditions (i.e., temperature and relative humidity) appear to affect significantly the performance of the sensors.

When compared with reference instruments, the CO and $NO_2$ electrochemical sensors provide measurements that exhibit increasing errors and decreasing correlations as temperature increases (from below 10 to above 30 °C) and RH decreases (from > 75 to below 30%). Interestingly, the performance of the sensors was affected irreversibly during the hot summer period, exhibiting different response before and after that, and resulting to a signal deterioration that was more than twice as that reported by the manufacturers. With the exception of the Alphasense $NO_2$ sensor, all LCSs exhibited measurement

uncertainties that were much higher, even at the beginning of our measurements, compared to those required for qualifying the sensors for indicative air quality measurements according to standard protocols. Overall, our results show that the response of all LCSs is strongly affected by the environmental conditions, warranting further investigations on how they are manufactured, calibrated and employed in the field.



## 1 Introduction

Air pollution accounts for more than 7 million premature deaths around the globe on an annual basis (Lelieveld et al., 2020), making it the fourth greatest overall risk factor for human health (Juginovic et al., 2021). Recognising this risk, the concentration of key pollutants has to be continuously monitored, especially in the urban environment, in order to ensure that those do not exceed certain limit values. Although wide networks of Air Quality (AQ) monitoring stations have been established in many cities around the globe for this reason, the spatial distribution of the observations is still limited for

providing detailed AQ mapping over urban agglomerates. This, in turn, limits our ability to effectively associate urban air pollution with potential human health effects, and to design effective mitigation strategies.

Building AQ monitoring networks that go far beyond the current state of art with respect to their spatial coverage is limited by their high capital, operational and maintenance costs. The city of Paris, for instance, operates 13 measuring sites (AIRPARIF, 2018), whereas in London AQ is monitored at around 100 locations (Greater London Authority, 2018). Although

these numbers are seemingly high, they correspond to a spatial coverage of the order of one station over a few tens of square kilometres. This limitation can be lifted by employing Low-Cost Sensors (LCSs) in dense observational networks that can enable quick and effective identification of pollution sources and determination of concentration gradients over specific areas.

According to the World Meteorological Organization (WMO), LCSs are defined as systems that are at least one order of magnitude less expensive compared to reference instruments (WMO, 2018). Electrochemical (EC) sensors, which fall into this

category, have received great attention in recent years as their low cost, compact size and response time (in the order of a minute) can enable their use in dense AQ monitoring networks and easy-to-carry-out mobile measurements. The low price of these sensors, however, comes at a cost in performance and in the quality of the data they provide compared to reference grade instruments, which can be attributed to a number of factors including cross-sensitivities (Lewis et al., 2015; Mead et al., 2013), low life expectancy or drift over time (Mead et al., 2013; Smith et al., 2017; Popoola et al., 2016; Hagan et al., 2018), low

sensitivities (Borrego et al., 2016), and variabilities related to environmental conditions (Mead et al., 2017; Jerrett et al., 2017; Spinelle et al., 2015; Spinelle et al., 2017; Cross et al., 2017; Pang et al., 2017; Aleixandre et al., 2012; Masson et al., 2015).

An increasing number of studies has investigated the performance of LCSs under different conditions (Hagan et al., 2018; Mead et al., 2013; Jerrett et al., 2017; Cross et al., 2017; Jiao et al., 2016; Bílek et al., 2021; Collier-Oxandale 2020). Closer comparison of the results from some of the above-mentioned studies, in which the performance of the same sensors (i.e.,

Alphasense B series) was tested but at different locations and conditions, shows that the variability of the environmental conditions plays a significant role provided the concentrations of the target gases are above their Limit of Detection (LoD) (Borrego et al., 2016; Mead et al., 2017; Cross et al., 2017; Bauerová et al., 2020; Karagulian et al., 2019). In a study carried out in Norway, during which the temperature and relative humidity (RH) varied from −0.7 to 23.3 °C and from 19 to 98%, respectively, the mean bias error (MBE) of the measurements reported by the CO and $NO_2$ LCSs when compared with

reference measurements were respectively −147.21 and 13.30 ppb, whereas the associated correlations ($R^2$) were 0.36 for the CO and 0.24 for the $NO_2$ LCS (Mead et al., 2017). In a similar study carried out in Portugal, where the temperature varied



from 15 to 30 °C and RH from 39 to 87%, the overall performance of the same sensors was better, exhibiting lower averaged MBEs (-0.025 ppb for CO and 7.1 ppb for $NO_2$) and higher $R^2$ (0.76 for CO and 0.58 for $NO_2$) (Borrego et al., 2016).

Operating temperature and RH are two of the most important factors affecting the performance of EC sensors mainly because they employ aqueous electrolytes containing sulphuric acid for the transfer of charges between the measuring and the reference electrodes. The concentration of sulphuric acid in the electrolyte is 5 M, which requires 60% RH at 20 °C in ambient air to be in equilibrium (Alphasense Application Note AAN106, 2013). Exposure of the sensors to RH levels less than 60% can in principle lead to evaporation of the solvent (i.e., water), whereas the opposite can happen at RH conditions above 60%. According to the manufacturer, the Alphasense EC sensors can provide meaningful measurements even when exposed to RH

values down to 15% or up to 90% at ambient temperature as long as they have enough time to equilibrate to the new conditions; i.e., several days. Ideally, when EC sensors are exposed over long periods to RH values far from 60%, they need to be recalibrated at the new RH conditions. However, considering that ambient RH is highly variable, and that it can change significantly within a few hours, using RH-dependent calibration turns out to be impractical.

It should be noted that field tests of LCSs reported in the literature were conducted at different locations, seasons, and

durations that range from 2 weeks (Borrego et al., 2016) to 4.5 months (Cross et al., 2017). This makes any direct comparison and further analysis of the data for understanding how the environmental conditions affect the performance of the LCSs highly challenging (Karagulian et al., 2019). In addition to the potential effects of the environmental conditions on the accuracy of LCSs, the long-term performance of LCSs can systematically vary over time as they can exhibit significant signal drifts (WMO, 2018; Jiao et al., 2016; Masson et al., 2015). To address these issues we urgently need systematic observations at different

locations and conditions over long periods of time.

In this work we compare yearlong measurements of the concentration of key gaseous pollutants (i.e., CO, $NO_2$, $O_3$ and $SO_2$) reported by low-cost EC sensors, manufactured by Alphasense and Winsen, and by reference-grade instruments. The measurements took place at a traffic station in the city of Nicosia, Cyprus, where the LCSs were exposed to highly variable gaseous concentrations, affected both by local and regional pollution sources, as well as variable environmental conditions.

The site is characterised by a high number of sunny days and long hot summers when the temperature and RH can reach values up to 50 °C and below 10%, respectively, providing a unique setting for testing the performance of these sensors under extreme conditions.

## 2 Methods

### 2.1 Low-cost gas sensors

EC sensors manufactured by Alphasense and Winsen EC were tested in this study. More specifically, we employed the analogue Alphasense B-series gas sensors (Models CO-B41, $NO_2$-B43F, OX-B431, and $SO_2$-B4) and the digital Winsen ZE12 model sensors configured by the manufacturer to measure the concentration of CO, $NO_2$, $O_3$ and $SO_2$.



The operating principle of EC sensors relies on redox reactions that take place on the surface of their sensing electrode, which is also referred to as the working electrode (WE). More specifically, the gas molecules reaching the surface of the WE

get either oxidised or reduced, consequently providing or capturing electrons to/from the system depending on their electronegativity. An EC cell is completed by the so-called counter electrode (CE) that balances the charge variabilities from the reactions at the surface of the WE. This charge balancing happens by transferring electrons through an electrolyte serving as the medium for transporting the charge carriers from one electrode to the other. The electrical signal (i.e., current) produced by this process is proportional to the concentration of the reducing/oxidising gas interacting with the WE. In some EC sensors,

a reference electrode (RE) is used to maintain the potential of the WE at a fixed value.

All EC LCSs employed in this work use the above-mentioned three-electrode configuration. The Alphasense sensors employ an additional electrode, namely the auxiliary electrode (AE), which has similar characteristics to those of the WE, but is incorporated in the cell in a way that it is not exposed to the target gas. A background current, caused by all other factors except the interaction of the WE with the target gas, is induced on the AE. It should be noted that the WE measures the

combined current, which is the sum of the background current and the current induced by the target gas when reacting with it. The AE current is then subtracted from the WE current, giving the compensated WE current (i.e., without the background signal) that corresponds solely to the interaction of the sensing electrode with the target gas.

The specifications of all the LCSs we employed are provided in Tables S1 in the supplement. For our tests the sensors were placed in two waterproof boxes – one containing the Alphasense and the other the Winsen LCSs (cf. Fig. 1a and 1b,

respectively) – with the sensing side facing outside as shown in Fig. 1c. The boxes, referred to as the AQ from this point onwards, were protected with a reflective cover to avoid direct exposure to sunlight, and attached on a railing within one-meter distance from the inlet of the reference instruments (cf. Fig. 1d). Both monitors were installed at the rooftop of the reference AQ monitoring station (see description further below) from October 2019 to December 2020, and operated continuously, with some interruptions during the high temperature seasons.

To run the AQ monitors and record the data from the LCSs we built an electronics circuit board (cf. Figures 1a and 1b) comprised of a microcontroller (Atmel, Model AVR Mega 2560), a Real Time Clock (RTC; Maxim, Model DS3231), a display and a Secure Digital (SD) card. For the Alphasense gas sensors that have a dual analogue output, an Analog to Digital Converter (ADC; Texas Instruments, Model ADS1115) was employed to convert their analogue voltage signals to digital values. The Winsen LCSs have a digital output in Universal Asynchronous Receiver-Transmitter (UART) protocol, so no conversion was

needed. The data from the LCSs were stored on the SD cards every 2 seconds for the Alphasense sensors and every 5 seconds for the Winsen sensors. The RTC, ADC and the display communicated with the microcontroller via an Inter-Integrated Circuit (I2C), whereas the SD card was accessed via Serial Peripheral Interface (SPI) protocol. Figure S1 in the supplement shows the schematic of the electronic circuit employed in the AQ monitors.



## 2.2 Measurement site

The site used for our measurements is a traffic AQ monitoring station in Nicosia, Cyprus, with latitude and longitude coordinates 35°9'7.20"N, 33°20'52.03"E. The station is located at a distance of ten metres from one of the main and busiest city avenues (cf. Fig. 2), which is typically congested during the morning and the afternoon rush hours. The site is considered a reference AQ station as the measurements follow the relevant EU Directives and the corresponding national laws defining the specifications that the employed instruments must meet, as well as the measurement procedures that must be followed,

which are also in accordance with international quality standards (Directive 2008/50/EC, 2008). The reference instruments used at the station together with their characteristics are listed in Table S2 of the supplement.

    We should highlight here that Cyprus is an island in the Eastern Mediterranean; a region that is a crossroad of polluted air masses coming from three continents (namely Europe, Asia and Africa; Lelieveld et al., 2002). Local sources contribute to the background AQ of the region, creating a highly interesting blend of air pollutants. In addition, the region is characterised by a

high number of sunny days with intense ultraviolet (UV) radiation, resulting to a strong atmospheric photochemical activity compared to other European countries (Vrekoussis et al., 2022). According to the national authority for regulatory monitoring, the main sources of CO observed on the island are related to local fossil fuel burning (i.e., energy production, transportation and central heating) having typical annual concentrations of more than 250 µg/m$^3$ (153 ppb) (DLI Annual Technical Report Air Quality, 2020). The primary sources of $NO_2$ and $SO_2$ emissions are also from fossil fuel burning, having typical mean

annual concentrations of more than 15 µg/m$^3$ (ca. 9 ppb) and 2 µg/m$^3$ (ca. 1 ppb), respectively (DLI Annual Technical Report Air Quality, 2020), whereas the annual mean concentrations of $O_3$ is ca. 70 µg/m$^3$ (ca. 35 ppb).

## 2.3 Calibration of the LCSs

    The Alphasense gas sensors employed in this work were accompanied by an Individual Sensor Board (ISB), which converts the signal from the WE and AE to voltages and facilitates data acquisition using ADCs. The electronic and zero offsets of the

WE and AE signals, as well as the sensitivity and temperature correction factors were also provided by the manufacturer for each sensor following laboratory calibration. These values were used to convert the measured signals to concentrations of pollutants by first calculating the corrected WE voltage value and then dividing by the sensor sensitivity. The manufacturer provides four different equations for the conversion of the raw signals from the WE and the AE to gas concentrations (expressed in ppb) for each sensor. The equations used in this work were the ones providing concentrations that are in better

agreement with those reported by the reference instruments. For the CO and $O_3$ sensors this is:

$$C = \frac{V_{WEc}}{S} = \frac{(V_{WEu} - V_{WEe}) - n_T * (V_{AEu} - V_{AEe})}{S}, \qquad (1)$$

where $V_{WEu}$ is the uncorrected raw WE output reported in mV, $V_{WEe}$ is the WE electronic offset on the ISB in mV, $n_T$ is a temperature dependent correction factor, $V_{AEu}$ is the uncorrected raw AE output in mV, $V_{AEe}$ is the AE electronic offset on ISB in mV, and S is the sensitivity in mV/ppb. For the $NO_2$ and the $SO_2$ sensors, the equation yielding concentration values

that compared better with those of the reference instrument is:





$$C = \frac{V_{WEc}}{S} = \frac{(V_{WEu} - V_{WEe}) - k_T * \left(\frac{V_{WEo}}{V_{AEo}}\right)(V_{AEu} - V_{AEe})}{S}. \qquad (2)$$

Here $V_{WEo}$ is the WE output in zero air (i.e., air without the target gas), $V_{AEo}$ is the AE output in zero air, and $k_T$ is the temperature dependent correction factor.

Temperature is used as input for the calculation of the correction factors $n_T$ and $k_T$ in Eq. (1) and (2). The equations for
determining $n_T$ and $k_T$ provided by the manufacturer for specific temperatures, ranging from -30 to 50 °C with 10 °C interval, are shown in the supplement (Table S3). It should be noted that for a more accurate calculation of the temperature correction factors in our analysis, we used linear (for CO) or spline (for $NO_2$, $O_3$ and $SO_2$) interpolations, as shown in Figure S2 in the supplement. The temperatures used as input for the calculation of the correction factors $n_T$ and $k_T$ are the ones measured by the reference thermometers at the station.

The Winsen ZE12 gas sensors are general-purpose sensors that can measure the concentration of CO, $SO_2$, $NO_2$ and $O_3$ based on the EC principle. In contrast to their Alphasense counterparts, these sensors do not come with a specific calibration equation to account for temperature variabilities. Instead, measurements from a built-in temperature sensor are used internally to capture respective variabilities, providing signal outputs that are directly expressed as concentrations (ppb).

## 2.4 Data Analysis

As stated above, the WE and AE outputs from the Alphasense LCSs were recorded every 2 seconds whereas the concentrations from the Winsen LCSs were recorded every 5 seconds. All LCS measurements were averaged over a period of an hour for direct comparison with the observations provided by the reference instruments. The effect of the environmental parameters (i.e. temperature and RH) on the performance of the sensors was assessed by dividing the whole dataset into different temperature and RH regimes.

The performance of the LCSs is evaluated by directly correlating and comparing the reported concentrations with measurements by the respective reference instruments to determine the associated errors. The parameters used to do so were the coefficient of determination ($R^2$), the Mean Bias Error (MBE), the Mean Relative Error (MRE), the Mean Absolute Error (MAE), and the Normalised Root Mean Square Error (NRMSE), defined respectively as follows:

$$R^2 = 1 - \frac{\sum_{i=1}^{N}(C_{LCS,i} - C_{ref,i})^2}{\sum_{i=1}^{N}(C_{LCS,i} - \overline{C_{ref}})^2} \qquad (3)$$

$$MRE = \frac{1}{N}\sum_{i=1}^{N}\left(\frac{(C_{LCS,i} - C_{ref,i})}{C_{ref,i}}\right) \times 100 \qquad (4)$$

$$MBE = \frac{1}{N}\sum_{i=1}^{N} C_{LCS,i} - C_{ref,i} \qquad (5)$$

$$MAE = \frac{1}{N}\sum_{i=1}^{N}\left|C_{LCS,i} - C_{ref,i}\right| \qquad (6)$$

$$NRMSE = \frac{1}{\overline{C_{LCS}}}\sqrt{\frac{\sum_{i=1}^{N}(C_{LCS,i} - C_{ref,i})^2}{N}} \qquad (7)$$



In all these equations, N is the total number of data points, whereas $C_{LCS,i}$ and $C_{ref,i}$ are the concentrations (expressed in ppb) measured respectively by the LCSs and the reference instruments at time $i$. To evaluate the degradation of sensors over time, we used the normalised MRE that is defined as follows:

$$NMRE = \frac{MRE_{end} - MRE_{bgn}}{MRE_{bgn}}. \tag{8}$$

Here $MRE_{bgn}$ and $MRE_{end}$ correspond to the MRE at the beginning and at the end of the measuring period, respectively.

In addition to assessing the performance of the sensors in terms of how well they correlate and agree with reference instruments, we determined whether the data they provide meet internationally accepted quality objectives. According to the 2008/50/EC AQ Directive (EC, 2008), the relative expanded uncertainty (REU) of the measurements can be used to assess the degree to which certain Data Quality Objectives are met for specific LCSs. REU is determined at fixed concentrations, which are typically the limit values set by directives and/or authorities, and are determined by collocating the LCSs with reference grade instruments at AQ monitoring stations. In general, the REU is defined as:

$$REU = \frac{2\sqrt{\frac{RSS}{N-2} + (\lambda - (\theta_1 - 1)^2) \times U_{ref}^2 + (\theta_0 + (\theta_1 - 1) \times C_{ref,i})^2}}{C_{ref,i}} \times 100\%. \tag{9}$$

Here N is the number of data pairs (i.e., time-tagged pairs of measurements from the LCSs and the reference instruments), $\lambda$ is the ratio of the variances of the LCSs over those of reference measurements which is typically assumed to be unity (Walker et al., 2020), $\theta_0$ and $\theta_1$ are estimated coefficients obtained via orthogonal regression between reference and LCS measurements (cf. Table S6 in the supplement), and $U_{ref}$ is the standard uncertainty of the reference measurements. RSS in Eq. (8) is the sum of the squared residuals determined by:

$$RSS = \sum_{i=1}^{N} (C_{LCS,i} - \theta_0 - \theta_1 \times C_{ref,i})^2. \tag{10}$$

A LCS can be used for indicative measurements if the REU determined at the limit value (or target value in the case of $O_3$) remains within 25% for CO, $NO_2$ and $SO_2$, or 30% for $O_3$ (Directive 2008/50/EC).

# 3 Results and Discussion

## 3.1. Statistics

Figure 3 provides a 2-month (over December 2019 and January 2020) snapshot of the hourly-averaged concentration measurements by the LCSs and the reference instruments. The CO and $NO_2$ sensors from both manufacturers are capable of qualitatively reproducing the reference measurements (cf. Figures 3a and 3b, respectively). It should be noted that an offset of ca. 400 ppb in the raw measurements from the Winsen CO LCS was observed throughout the entire period of our study, and that this value has been subtracted from the reported data to facilitate comparison with the measurements by the respective reference instrument. In contrast to the case of CO and $NO_2$, the $SO_2$ and $O_3$ concentrations measured by the LCSs exhibit a difference of two to three orders of magnitude compared to the reference measurements. For display purposes, the $SO_2$ LCS



measurements shown in Fig. 3 were divided by 10; the results discussed in the rest of the paper, however, correspond to the real recorded values.

Aggregated statistics from all the measurements and for each sensor during the entire testing period are provided in Table 1. Overall, the CO measurements by the LCSs from both manufacturers show good agreement with those reported by the respective reference instruments (cf. mean values in Table 1). Both CO sensors capture the temporal concentration variability attributed primarily to the morning and afternoon traffic pollution. The Alphasense CO LCS exhibits the lowest MRE, with a value of -5.1%, and the highest correlation ($R^2 = 0.56$) compared to the respective Winsen LCS (MRE = 35.1%, $R^2 = 0.25$).

The measurements recorded by the $NO_2$ LCSs from both manufacturers followed similar temporal patterns to those of the reference instrument, exhibiting moderate, but similar, correlations (i.e., $R^2 = 0.34$ for Alphasense and $R^2 = 0.29$ for Winsen). Overall, both LCSs overestimated the $NO_2$ concentrations, with the Alphasense $NO_2$ sensor exhibiting better performance compared to that of Winsen as indicated by the MRE (61.0% against 104.9%, respectively) and MAE (8.6 against 15.1 ppb, respectively) shown in Table 1.

The Alphasense $O_3$ sensor overestimated significantly the absolute reference concentrations, exhibiting high MRE (901.1%) and MAE (52.4 ppb) values. On the other hand, the Winsen $O_3$ sensor mainly underestimated the actual concentration, exhibiting a significantly lower MRE (44.0%) and MAE (-10.6 ppb) compared to its Alphasense counterpart. The reported correlation coefficients of both $O_3$ sensors were among the lowest of all the LCSs tested, showing rather poor responses ($R^2$ was 0.18 for the Alphasense and 0.06 for the Winsen sensor).

The performance of the $SO_2$ sensors was the poorest compared to those of the rest of the LCSs mainly because the concentrations at the monitoring station, as reported by the reference instruments, were quite low (ranging between 2 and 4 ppb during the study period) and always below the LoD of the LCSs: i.e., 5 ppb for the Alphasense and 15 ppb for the Winsen sensor.

Figure 4 shows the correlation of the measurements recorded by the Alphasense (a-d) and by the Winsen (e-h) LCSs with those provided by the respective reference instruments. The associated errors between the LCS and reference instrument measurements are provided in the supplement (cf. Fig. S3). The performance of LCSs appears to degrade from left to right in Fig. 4, with the CO and $NO_2$ exhibiting the best performance, whereas $O_3$ and $SO_2$ the worst.

Two distinct linear data clusters are observed for the Alphasense CO sensor (cf. Fig. 4a), which can be attributed to the degradation of the sensor performance from its exposure to the high temperatures and low RH values during the summer period. This is more clearly shown in Fig. S4a in the supplement, which depicts the same data tagged by the time that the measurements were recorded. We should note here that the temperatures and RH conditions that the LCSs were exposed to during the entire measuring campaign ranged respectively from 3 to 43 °C and from < 10 to 97%. As discussed in Sect. 1, exposing the EC LCSs to extreme temperature and RH conditions can cause irreversible changes of their electrolyte solution, leading to a behaviour that is noticeably different before and after that. The phenomenon is also observed for the Winsen CO LCS (cf. Fig. S4b in the supplement), but to a less extend mainly due to the limited data acquired after summer with that sensor. The same is not evident for the rest of the sensors. However, we should note here that correlation with reference





instruments was not as strong in those cases, potentially resulting in the small extent of data clustering before and after the summer period, and that the amount of useful measurements used for the analysis was less than that of the CO Alphasense sensor (cf. Table 1).

The data provided in Table 1 and Fig. 4 indicate that the $O_3$ and $SO_2$ LCSs by both manufacturers exhibit extremely high MREs and/or low correlations against the respective reference instruments. The high errors of the Alphasense $O_3$ measurement can be associated to interferences with $NO_2$. This is in fact taken into account, as the signal from the $NO_2$ LCS (mV) is subtracted from the $O_3$ sensor signal following the guidelines by the manufacturer (cf. first term of Eq. (1)). Doing so, however, leads to a measurement error that is higher compared to the case when the correction is not applied (cf. Fig. S6), most likely

due to error propagation from the $NO_2$ sensors signal to the reported concentrations of $O_3$. We should note that no correction is provided by Alphasense for the case of the $NO_2$ sensor. For the case of the $SO_2$ LCSs, as indicated above, most of the measurements fall below the threshold of their LoD (cf. Table S1). Considering the above, the performance of the $O_3$ and $SO_2$ sensors was not assessed further, and thus we will not discuss the respective measurements from this point onwards.

### 3.2. Effect of temperature

Figures 5 and 6 show the correlation between the CO or the $NO_2$ measurements recorded by the Alphasense (Figures 5a-d and 6a-d) and the Winsen (Figures 5e-h and 6e-h) LCSs, with the respective measurements provided by the reference instruments at four temperature ranges: T < 10 °C, 10-20 °C, 20-30 °C and > 30 °C. Statistics and parameters of linear regression fittings to the data at each temperature range are provided in Table 2.

        Overall, the trend is characterised by a decreasing correlation between LCS and reference measurements as the temperature

increases. More specifically, the Alphasense CO sensor (cf. Fig. 5a-5d) correlates well with the reference instrument at temperatures below 20 °C, than at higher temperatures (e.g., $R^2$ is 0.69 for T 10-20 °C and 0.05 for T > 30 °C; cf. Table 2). In a similar manner, the Winsen CO sensor (cf. Fig. 5e-5h) exhibits strong correlation with reference measurements at temperatures below 10 °C (e.g., $R^2$ = 0.69; cf. Table 2), but deteriorates at higher temperatures, exhibiting anti-correlation above 20 °C (cf. Table 2).

Similarly to the CO LCSs, the $NO_2$ sensors exhibit the highest correlation with reference measurements at temperatures below 10 °C, and the lowest above 30 °C. More specifically, the Alphasense $NO_2$ sensor (cf. Fig. 6a-d) exhibits high correlations with the reference instrument ($R^2$ = 0.80) below 10 °C, which is similar to that of the Winsen $NO_2$ sensor (cf. Fig. 6e-h; $R^2$ = 0.59). Above this temperature, the sensor performance deteriorates at the extent that it becomes incapable of capturing the variability of the $NO_2$ concentrations (cf. Fig. 6f-6h). In addition to that, $NO_2$ LCSs can have (according to

Alphasense) interferences from $O_3$ (Li et al., 2021), which is typically found in high concentrations during high UV radiation periods of the summer months in Cyprus (Vrekoussis et al., 2022), but this is not accounted for by the manufacturer as discussed also in the previous section.

        The extreme temperatures that the LCSs have been exposed to over the summer period, have significantly affected their performance during the two cold periods of our tests. This is more pronounced for the CO sensors as reflected by the two




distinct data clusters in Fig. 5a and 5e, as well as in Fig. S4 in the supplement. As discussed above, this can be attributed to changes of electrolyte water content during high temperature seasons as in the case of the Alphasense CO LCS (cf. Sect. 3.1). It should be noted that even though the sensors were not operating during the summer months, they were exposed to ambient temperatures (up to 45 °C), while their main body containing the electrolyte was at even higher values (i.e., > 50 °C) during midday, as a result of heat built up within the cases of the AQ monitors.

**3.3. Effect of relative humidity**

Figures 7 and 8 show respectively the correlation between CO and $NO_2$ measurements recorded by the Alphasense and the Winsen LCSs, and those provided by the respective reference instruments at four RH ranges: < 30%, 30-55%, 55-75% and > 75%. Statistics and parameters of linear regression fittings at each RH range are provided in Table 3.

The correlation between the LCSs and the reference instruments increases with increasing RH. The Alphasense CO sensor
(cf. Fig. 7a-7d) correlates better with the reference measurements at RH values above 75% (i.e., $R^2$ up to 0.70; cf. Table 3), than at drier conditions (i.e., $R^2 = 0.20$ for RH < 30%; cf. Table 3). In a similar manner, the Winsen CO sensor (cf. Fig. 7e-7h) exhibits strong correlation with reference measurements at RH values above 75% (i.e., $R^2 = 0.57$; cf. Table 3), but deteriorates at lower RH values.

The $NO_2$ LCSs from the two manufacturers exhibit the highest correlation with reference measurements at RH > 75%, but
decrease substantially at lower RH values. More specifically, the Alphasense $NO_2$ sensor (cf. Fig. 8a-d) exhibits higher correlations with the reference instrument ($R^2 = 0.56$) at RH > 75%, compared to lower RH levels (i.e., $R^2 = 0.31$ at RH < 30%). The measurements by the Winsen $NO_2$ LCS (cf. Fig. 8e-h) exhibited similar correlations with its Alphasense counterpart at RH > 75%, but its performance deteriorates at RH values below 75%, until it becomes incapable of capturing the variability of the $NO_2$ concentrations (cf. Fig. 8e-6g). Since low RH is associated with elevated temperatures, and vice versa (cf. Fig. S5
in the supplement), it appears that the degraded performance noted at high temperatures for the CO and $NO_2$ sensors from both manufacturers corresponds to that observed at the low RH conditions (cf. correlations provided in Fig. 5d, 5h, and 6d, 6h in correspondence with those in Fig. 7a, 7e and 8a, 8e).

The Alphasense EC gas sensors are calibrated in the lab at a fixed RH of 60%, but according to the manufacturer they can operate in the RH range of 15 to 90%. Long exposures of the sensors to an environment with a RH below or above 60% can
affect their performance due to evaporation or condensation, respectively, of water from/to the electrolyte (Alphasense Application Note AAN106, 2013). In principle, re-calibration of these sensors would be appropriate provided that they equilibrate to the new RH conditions. In practice, however, this is not so practical to do in the field because the period required for the sensor to reach equilibrium is much longer (ranging from a few days to almost a month) compared to the daily variability of ambient RH.

As discussed in Sect. 3.2, even though the sensors were not operating during the summer months, their main body, which was within the AQ monitors, was exposed to extreme conditions (temperatures > 50 °C and RH < 10%). In addition to that, the sensors were exposed to significant diurnal variations; from 20 °C during the night to 40 °C during the day, with respective



values of RH ranging from 20 to 80% (cf. Fig. S5 in the supplement), and thus they never had enough time to equilibrate to any RH. These conditions had an impact on the performance of the sensors which was not only temporary (i.e., during the
summer period), but persisted even after summer as discussed above (cf. Sect. 3.1).

### 3.4. Degradation of Low Cost EC Sensors

Figures 9 and 10 show how the measurements from the CO and $NO_2$ LCSs respectively compare with those from reference instruments over 6 consecutive days at the beginning and at the end of the measuring period. Table 4 summarises the results providing the median MBE, MRE, and NMRE, along with the temperature, RH and reference concentration corresponding to
the two periods.

Evidently, all LCSs exhibit degradation of their performance over time, expressed as an increase of the median MBE, in terms of absolute values, after the period of one year (cf. Figures 9-10). Similarly, an increase of the MRE values, again in absolute terms, is also observed. For the Alphasense CO the MRE changes from almost 32 to less than -52% from the start to the end of the field campaign, yielding an NMRE of -2.6, whereas for the Alphasense $NO_2$ sensors the MRE changes from
almost 8 to around 23%, with a NMRE of 1.9. These results are in line with those reported by Li et al. (2021), who showed that most of the Alphasense $NO_2$ sensors they tested exhibited significant deterioration of their performance after long-term (200 to 400 days) deployment. The Winsen sensors exhibited similar degradation in performance, with NMRE values of 2.0 for the CO and 1.3 for the $NO_2$ sensors. We should note here that to make a fair comparison of the performance of each sensor at the beginning and the end of the measuring campaign, we made an effort to select measurements corresponding to similar
temperature and RH conditions during the two periods. This was feasible for the datasets of the Alphasense sensors but not so easy for the Winsen, as those were not always functional after the summer period, yielding significantly less measurements during that period as indicated in Table 1 and reflected by the high temperature differences in the two periods analysed here (cf. Table 4).

According to specifications provided by the manufacturers, the lifespan of EC LCSs ranges from 2 to 3 years, and
specifically for the Alphasense sensors the signal deterioration should be within less than 50% over a period of 24 months (cf. Table 1 in the supplement and references therein). Assessing that based on field measurements, however, is not so straightforward due to simultaneous variabilities in both the concentration of the target gases and the meteorological conditions over time. To overcome this problem, we selected two 2-month periods, one at the beginning (October-November 2019) and one at the end (November-December 2020) of our field study, constraining the temperature below 20 °C and RH above 55%,
while the reference concentration for the selected data during this period was bracketed within ±10% from the median annual reference concentration (i.e., 467.7 ppb for CO and 19.4 ppb for $NO_2$). By comparing the resulting Alphasense LCSs median raw signals (i.e., $V_{WEu} - V_{AEu}$) at the beginning and at the end of the measuring period (cf. Fig. 11), a total drift of 86.1 mV is observed for CO and 17.2 mV for $NO_2$ , after one year of operation, corresponding to drifts that are respectively 53 and 68% from the initial signal of the sensors. These values significantly exceed the estimated sensor drifts in the specifications,
indicating that they increase with a rate that is more than double to that suggested by the manufacturer. This discrepancy can





very well be attributed to the high temperatures and low RHs encountered during our field evaluation, which in some cases exceeded the operational limits set by the manufacturer.

## 3.5. Compliance with EU guideline standards and data quality objectives

As already reported in the literature, data from LCSs do not meet the quality objectives for use in regulatory observations
stated for instance by the associate EU directive (Peltier et al., 2020; Directive 2008/50/EC, 2008). A key question, however, is whether they can still be used to provide indicative measurements, which require less accuracy and can complement the regulatory observations by increasing the spatiotemporal resolution of AQ measurements. To address this question, the sensors are required to (Directive 2008/50/EC, 2008): 1. have adequate LoD for measuring concentrations below the target/limit value and even lower than the assessment threshold concentrations set by the EU guidelines, 2. provide measurements that fulfil the
required minimum data capture and time coverage for indicative measurements, and 3. provide measurements that are within the minimum REU, which according to the EU legislation should remain within 25% for the two gases that we further analyse here (i.e., CO and $NO_2$; cf. Sect. S.6. in the supplement for more details).

Both CO and $NO_2$ sensors have sufficiently low LoDs for measuring concentrations in the order of the EU threshold concentrations and even significantly below that. They can also provide rich datasets, achieving data capture and time coverage
that is well above the set standard thresholds. Considering the above, the first two requirements mentioned in the previous paragraph are fulfilled, leaving the last as the main limitation for qualifying these LCSs for indicative measurements.

Figure 12 shows the REU in the measurements provided by the Alphasense (Fig. 12a-b) and Winsen (Fig. 12c-d) CO and $NO_2$ LCSs as a function of the respective reference concentrations. We should note here that the data were averaged over 8 hours for the CO and over 1 hour for $NO_2$ concentrations, following the EU standard for the limit values; 8600 ppb over 8
hours for CO and 105 ppb over 1 hour for $NO_2$ (Directive 2008/50/EC, 2008; cf. supplement Table S4). As shown, the sensors did not comply with the required REU limit, even at the beginning of the measuring period when they were not affected by the summer extreme conditions. We should note here that the EU limit value for CO concentrations is much higher compared to those measured at the AQ monitoring station we used in Nicosia, whereas for $NO_2$ they approached but never exceeded it. Nevertheless, the REU values for the two CO sensors appear to reach a plateau as the concentration of the target gas increases,
which was higher for the period after the summer, and even if they are extrapolated to the limit values they would not reach the required 25% threshold. The same is true for the Winsen but not for the Alphasense $NO_2$ sensor which appears to reach this threshold at concentrations higher than ca. 70 ppb (cf. Fig. 12b), making it the only sensors that can qualify for indicative measurements according to the EU directive.

## 4 Conclusions

We have investigated the performance of low-cost electrochemical sensors provided by two manufacturers, namely Alphasense and Winsen, for measuring the concentrations of CO, $NO_2$, $O_3$ and $SO_2$. To do so we carried out yearlong measurements at a



traffic AQ monitoring station in Nicosia, Cyprus, where the LCSs were collocated with reference instruments and exposed to highly variable environmental conditions, with extremely high temperatures (up to 45 °C) and low relative humidity conditions (below 10%) during the summer period.

Among the different LCSs tested, the CO sensors exhibited the best overall performance reporting measurements that had high correlation with, and low deviation from, those reported by reference instruments throughout the entire testing period. The $NO_2$ gas sensors from both manufacturers also exhibited good performance, having satisfactory correlations and deviations when compared to the respective reference instrument. In contrast, the performance of the $O_3$ sensors were not satisfactory, while the $SO_2$ sensors were not possible to evaluate because the ambient concentrations were below their LoDs.

Our measurements demonstrate that the performance of the CO and $NO_2$ sensors from both manufactures is affected by exposure to temperatures above 25 °C and RH levels below 55%. This is most probably caused by the evaporation of water from the sensor electrolyte, which in turn affects the currents induced by the interaction of the target gases with the working electrodes through the sensors' cell. After a prolonged use under extreme conditions (i.e., temperatures above 40 °C and relative humidity conditions below 10%) the sensors exhibited high errors, and in some cases technical malfunctions, making

them practically unusable. Although the sensors regained their operability (i.e., being able to provide uninterrupted data) as the conditions became milder, their performance did not fully recover. Exposure of the sensors to these extreme conditions appeared to cause a drift of more than 50% over a period of a year, which is more than twice as fast compared to that indicated by the manufacturers.

Although the LCSs tested in this work fulfilled two of the three criteria (i.e. adequate LoD, measurements that fulfil the

required minimum data capture and time coverage) that qualifies them for indicative measurements according to EU standards, they fail the third criterion of exhibiting REU values below 25%. The only exception to this generalisation was the Alphasense $NO_2$ sensor which exhibited REU values close to 25% for concentrations above 70 ppb, which are below the hourly limit $NO_2$ value, making it the only LCS that can be used for indicative measurements according to the EU directive.

**Code/Data availability**

All raw data can be provided by the corresponding authors upon request.

**Author contribution**

RP drafted the manuscript that was reviewed and edited by SB and GB; MC and NH designed and developed the monitors; RP and MD performed the measurements; MS, ES and CS provided the reference station data; RP analysed the data. The research was conceptualised by SB and GB.



**Competing interests**

The authors declare that they have no conflict of interest.

**Acknowledgements**

1. EMME-CARE: This project has received funding from the European Union's Horizon 2020 research and innovation programme under grant agreement No. 856612 and the Cyprus Government.

2. AQ-SERVE: The Project INTEGRATED/0916/0016 (AQ-SERVE) is co-financed by the European Regional Development Fund and the Republic of Cyprus through the Research and Innovation Foundation.

3. ACCEPT: The project "ACCEPT" (Prot. No: LOCALDEV-0008) is co-financed by the Financial Mechanism of Norway (85%) and the Republic of Cyprus (15%) in the framework of the programming period 2014 - 2021.

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



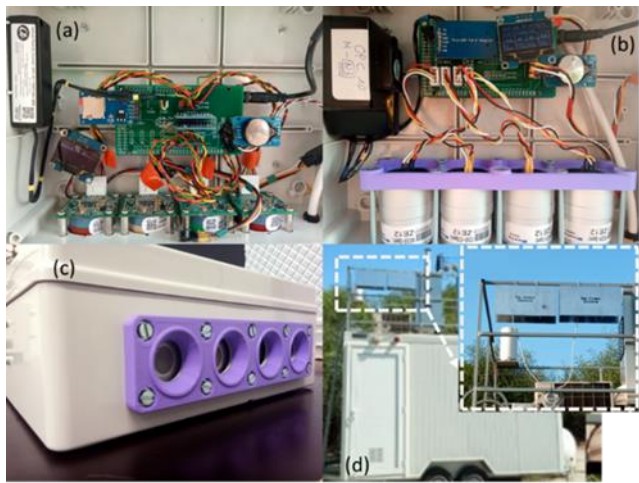

**Figure 1: Images showing the layout of the AQ monitors housing the Alphasense (a) and the Winsen (b) LCSs, the side of the AQ monitors at which the sensing elements of the LCSs were exposed to the ambient air (c), and the place that the AQ monitors were installed at the top of the reference station (d).**

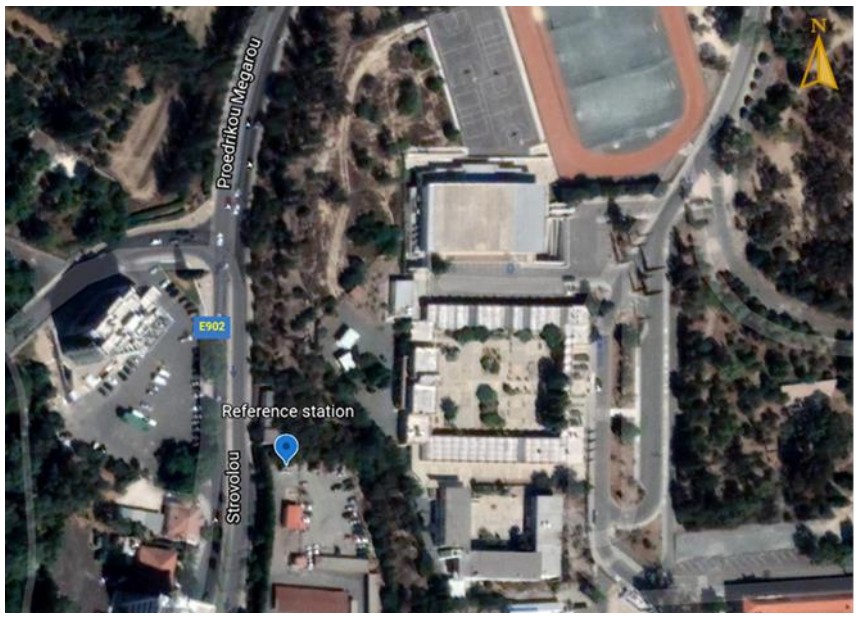

**Figure 2: Satellite image showing the location of the traffic AQ monitoring station in Nicosia where the measurements were carried**
**out. The station is located ca. 10 m from Strovolou avenue, which is a rather busy road especially during the morning and evening rush hours. Map data are from © Google, DigitalGlobe.**



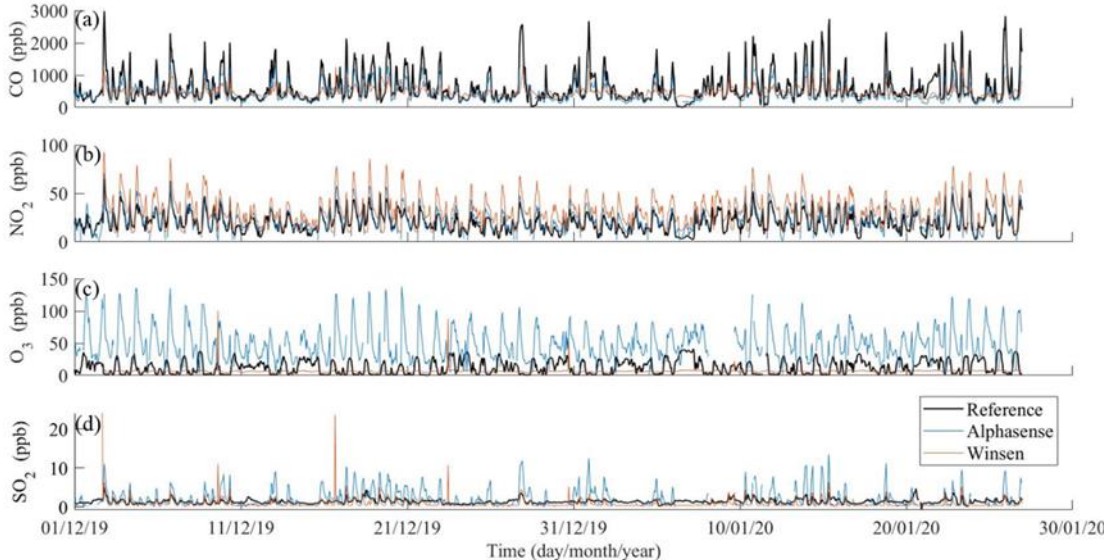

**Figure 3: Time series of hourly averaged concentrations measured by the Alphasense and Winsen EC gas sensors and the reference instruments from 1 December 2019 to 26 January 2020 for CO (a), NO$_2$, (b), O$_3$ (c) and SO$_2$ (d). We should note here that the data from the LCSs were determined through the calibration equations discussed in Sect. 2.4 (for the Alphasense sensors) or provided directly by the sensors (for the Winsen sensors). Exception to this are the data from the Winsen CO LCS, for which we subtracted 400 ppb from all the measurements, and the SO$_2$ LCSs from both manufacturers were the signal was divided by a factor of 10 for illustration purposes.**

**Table 1: Summary statistics, including the number of useful measurements for each LCS (N) and the mean concentration of hourly averaged gaseous concentrations measured by the Alphasense and Winsen LCSs and the reference instruments for the entire study period, together with the Standard Deviation (SD) for each case, as well the associated values of the MRE, MBE, MAE and R$^2$. The LoD values for each LCS are also provided for comparison.**

| Type of sensor | LCS N | LCS Mean ± SD | Reference Mean ± SD | MRE (%) | MBE (ppb) | MAE (ppb) | R$^2$ | LoD (ppb) |
|---|---|---|---|---|---|---|---|---|
| **Alphasense CO** | 5828 | 432.8 ± 277.0 | 667.4 ± 665.9 | -5.1 | -234.7 | 304.8 | 0.56 | 4 |
| **Winsen CO[1]** | 3784 | 419.6 ± 156.3 | 531.1 ± 412.0 | 35.1 | -111.3 | 256.3 | 0.25 | 130 |
| **Alphasense NO$_2$** | 3990 | 23.8 ± 13.3 | 20.3 ± 13.7 | 61.0 | 3.49 | 8.6 | 0.34 | 15 |
| **Winsen NO$_2$** | 3900 | 31.2 ± 19.3 | 19.5 ± 12.2 | 104.9 | 11.7 | 15.1 | 0.29 | 15 |
| **Alphasense O$_3$** | 5048 | 74.6 ± 55.8 | 27.5 ± 23.8 | 901.1 | 47.1 | 52.4 | 0.18 | 15 |
| **Winsen O$_3$** | 3879 | 10.2 ± 23.4 | 20.8 ± 18.3 | 44.0 | -10.6 | 16.5 | 0.06 | 15 |
| **Alphasense SO$_2$** | 3864 | 23.3 ± 22.8 | 1.8 ± 1.0 | 1354.4 | 21.5 | 21.6 | 0.12 | 5 |
| **Winsen SO$_2$** | 3888 | 7.1 ± 7.6 | 1.5 ± 0.65 | 389.5 | 5.6 | 5.6 | 0.17 | 15 |

---

[1] Measurements recorded by the Winsen CO sensors were reduced by 400 ppb in order to better match the measurements from the reference instruments. The Winsen CO statistics provided in this Table correspond to the corrected data.



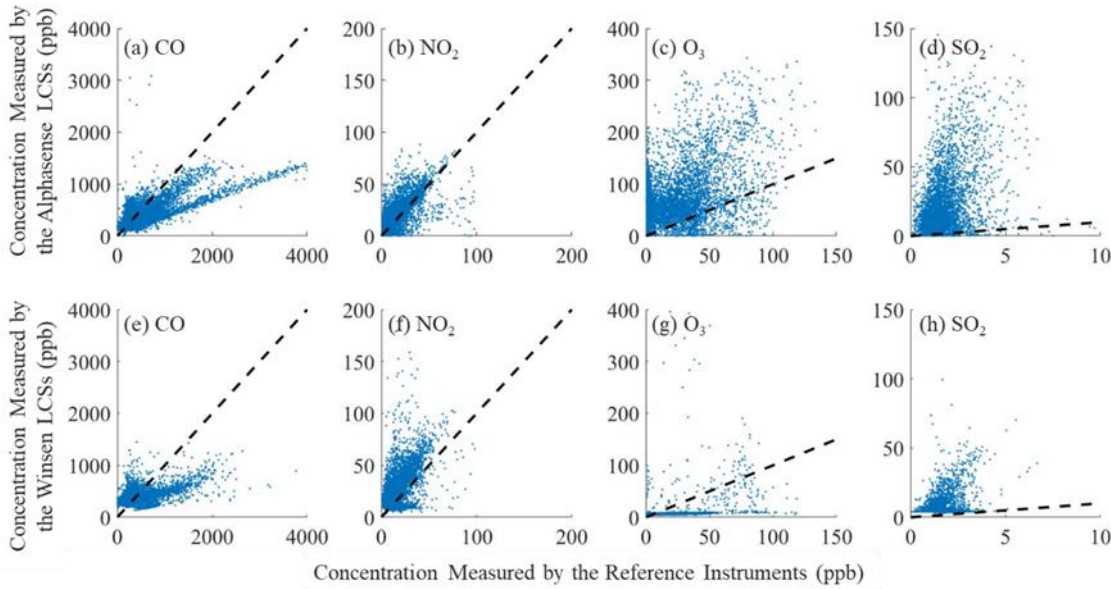


**Figure 4: Correlation between the measurements recorded by the Alphasense (a-d) and Winsen (e-h) LCSs and those provided by the respective reference instruments. The black dashed lines indicate the 1:1 line. $O_3$ and $SO_2$ sensors of both manufacturers do not have the same x-y axis scales due to the big difference between reference and LCSs measurements recorded.**

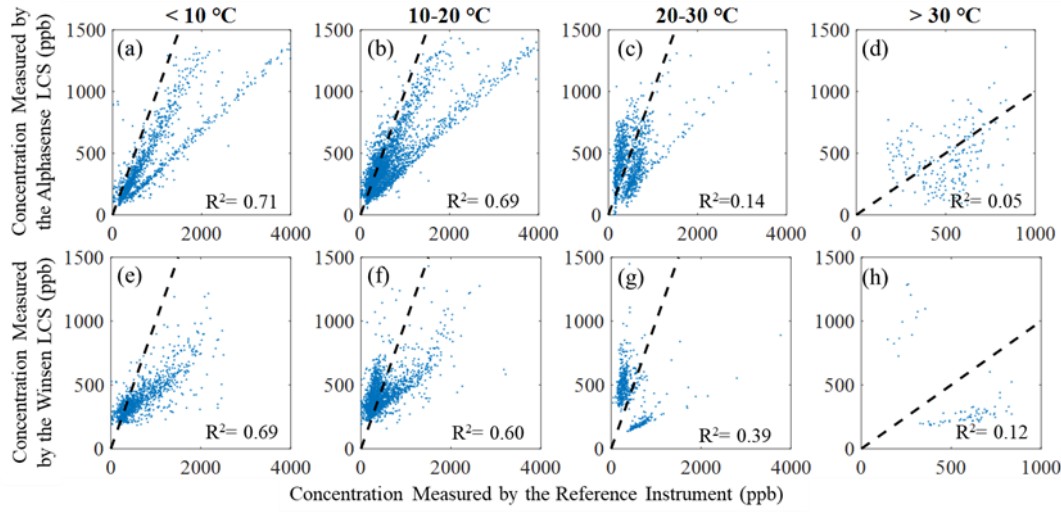


**Figure 5: Correlation between the CO measurements recorded by the Alphasense (a-d) and the Winsen (e-h) LCSs, and those provided by the respective reference instrument at four different temperature ranges. The 1:1 lines for each subplot are depicted as dashed lines.**




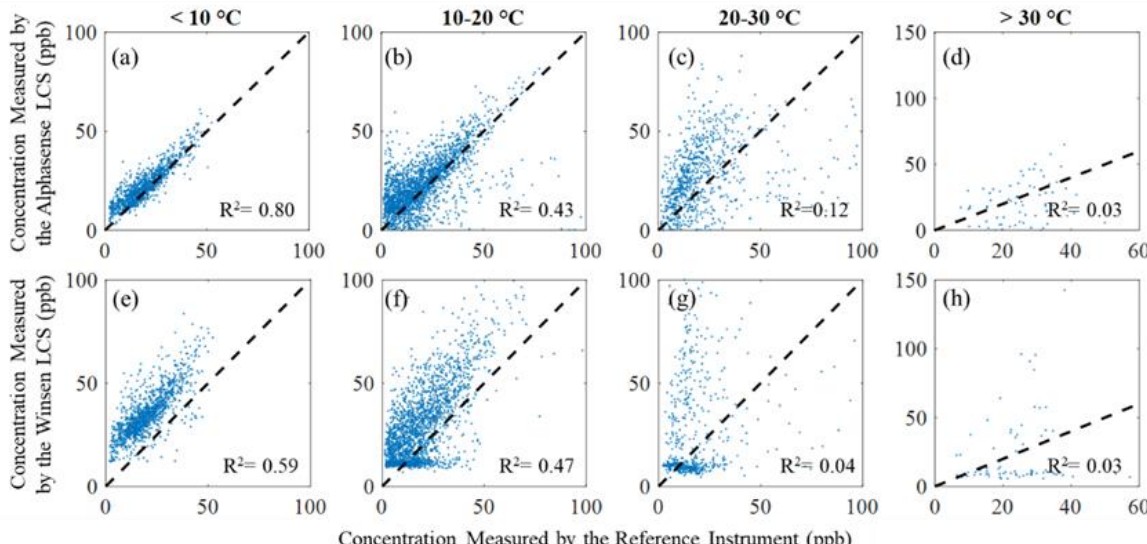


**Figure 6: Correlation between the NO$_2$ measurements recorded by the Alphasense (a-d) and the Winsen (e-h) LCSs and those provided by the respective reference instrument at four different temperature ranges. The 1:1 lines for each subplot are depicted as dashed lines.**

**Table 2: Summary statistics including the number of useful measurements (N) and the MRE for each LCS at four different temperature ranges over the entire study period. Linear regression parameters including NRMSE, slope, intercept and R$^2$ are also provided for each temperature range.**

| Type of sensor | Temperature range (°C) | N | MRE (%) | Linear regression | | | |
|---|---|---|---|---|---|---|---|
| | | | | NRMSE | Slope | Intercept (ppb) | R$^2$ |
| Alphasense CO | < 10 | 1301 | -29.0 | 0.39 | 0.33 | 164.4 | 0.71 |
| | 10-20 | 3154 | -4.4 | 0.35 | 0.32 | 204.7 | 0.69 |
| | 20-30 | 1170 | 17.2 | 0.50 | 0.25 | 318.5 | 0.14 |
| | >30 | 203 | 9.4 | 0.66 | 0.41 | 278.3 | 0.05 |
| Winsen CO | < 10 | 1163 | -9.5 | 0.42 | 0.25 | 230.4 | 0.69 |
| | 10-20 | 2034 | 55.5 | 0.54 | 0.23 | 312.8 | 0.60 |
| | 20-30 | 553 | 54.7 | 0.76 | -0.09 | 491.3 | 0.39 |
| | > 30 | 72 | 7.7 | 1.01 | -1.13 | 1014.2 | 0.12 |
| Alphasense NO$_2$ | < 10 | 1057 | 33.7 | 0.18 | 0.84 | 6.1 | 0.80 |
| | 10-20 | 2094 | 72.5 | 0.43 | 0.61 | 10.7 | 0.43 |
| | 20-30 | 767 | 73.9 | 0.56 | 0.35 | 20.3 | 0.12 |
| | > 30 | 72 | -8.8 | 0.70 | 0.23 | 15.3 | 0.03 |
| Winsen NO$_2$ | < 10 | 1180 | 111.6 | 0.20 | 0.87 | 17.6 | 0.59 |
| | 10-20 | 2100 | 103.4 | 0.45 | 0.97 | 10.8 | 0.47 |
| | 20-30 | 561 | 109.4 | 1.00 | 0.45 | 21.8 | 0.04 |
| | > 30 | 72 | 8.7 | 1.10 | 0.44 | 13.4 | 0.03 |



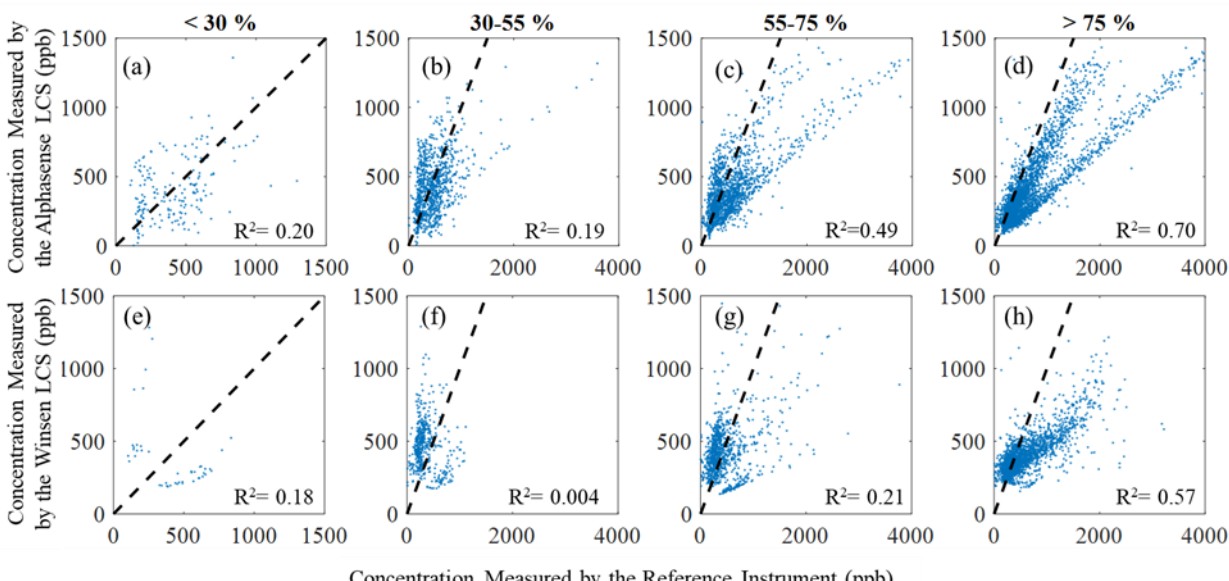

**Figure 7: Correlation between CO measurements recorded by the Alphasense (a-d) and the Winsen (e-h) LCSs and those provided by the respective reference instrument at four different RH ranges. The 1:1 lines for each subplot are depicted as dashed lines.**

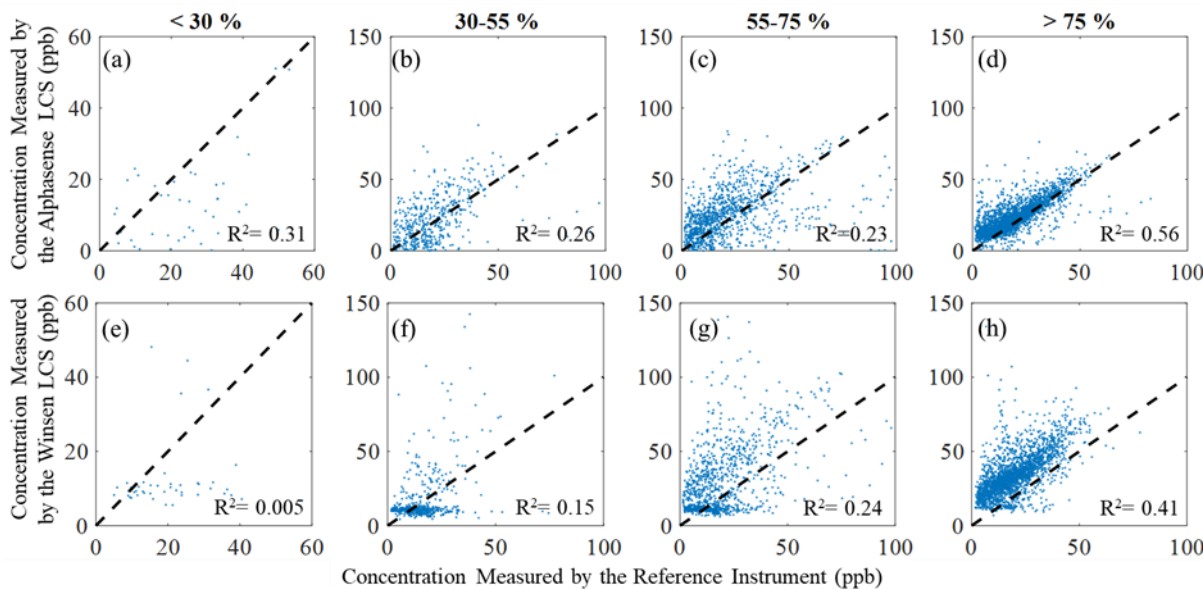

**Figure 8: Correlation between the NO$_2$ measurements recorded by the Alphasense (a-d) and the Winsen (e-h) LCSs and those provided by the respective reference instrument at four different RH ranges. The 1:1 lines for each subplot are depicted as dashed lines.**




**Table 3: Summary statistics including the number of useful measurements (N) and the MRE for each LCS at four different RH ranges over the entire study period. Linear regression parameters including NRMSE, slope, intercept and $R^2$ are also provided for each RH range.**

| Type of sensor | RH range (%) | N | MRE (%) | Linear regression fitting | | | |
|---|---|---|---|---|---|---|---|
| | | | | NRMSE | Slope | Intercept (ppb) | $R^2$ |
| **Alphasense CO** | < 30 | 189 | 18.8 | 0.48 | 0.44 | 219.6 | 0.20 |
| | 30-55 | 1011 | 21.3 | 0.49 | 0.30 | 289.4 | 0.19 |
| | 55-75 | 1632 | 4.2 | 0.44 | 0.32 | 221.7 | 0.49 |
| | > 75 | 2996 | -20.6 | 0.37 | 0.32 | 191.8 | 0.70 |
| **Winsen CO** | < 30 | 43 | 56.1 | 0.61 | -0.55 | 616.9 | 0.18 |
| | 30-55 | 509 | 92.6 | 0.36 | -0.05 | 490.5 | 0.004 |
| | 55-75 | 1023 | 67.0 | 0.38 | 0.20 | 336.0 | 0.21 |
| | > 75 | 2250 | 6.7 | 0.23 | 0.22 | 266.0 | 0.57 |
| **Alphasense NO₂** | < 30 | 43 | -31.3 | 0.79 | 0.40 | 3.7 | 0.31 |
| | 30-55 | 471 | 49.4 | 0.63 | 0.58 | 10.3 | 0.26 |
| | 55-75 | 1090 | 76.9 | 0.56 | 0.45 | 15.8 | 0.23 |
| | > 75 | 2386 | 57.8 | 0.30 | 0.71 | 9.6 | 0.56 |
| **Winsen NO₂** | < 30 | 43 | 26.4 | 0.79 | 0.06 | 11.06 | 0.005 |
| | 30-55 | 528 | 41.3 | 0.99 | 0.73 | 6.08 | 0.15 |
| | 55-75 | 1046 | 115.3 | 0.70 | 0.80 | 14.9 | 0.24 |
| | > 75 | 2299 | 117.3 | 0.32 | 0.85 | 17.8 | 0.41 |

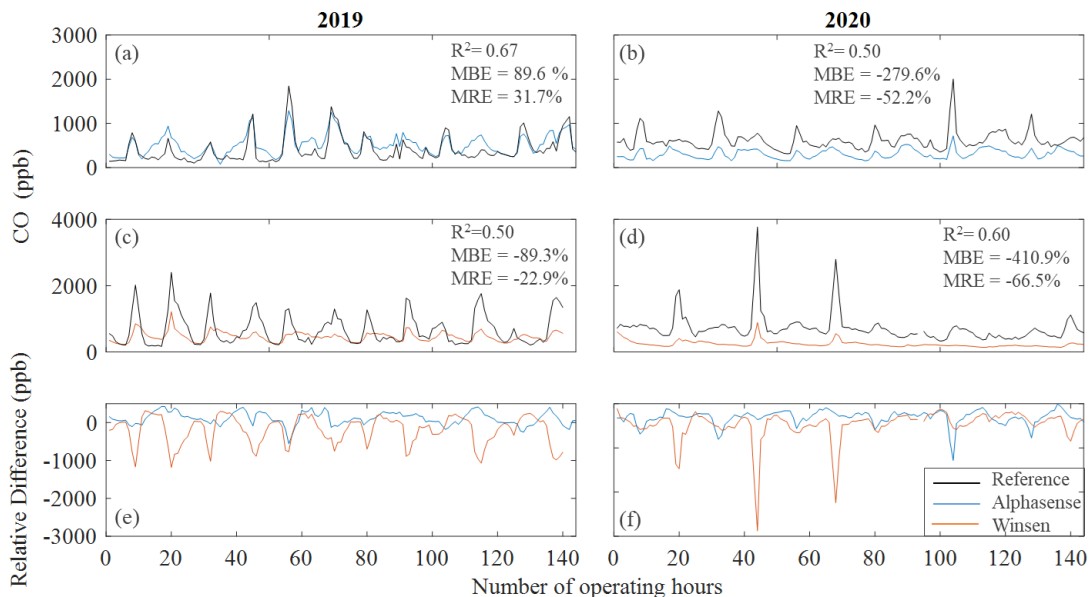

**Figure 9: Time series of hourly-averaged concentrations measured by the Alphasense (a-b) and Winsen (c-d) CO LCSs over six-consecutive days at the beginning (left column; October 2019 for Alphasense and November 2019 for Winsen) and at the end (right column; October 2020 for Alphasense and September 2020 for Winsen) of our measuring period. Last row shows the concentration actual difference between CO LCSs and reference instrument in 2019 (e) and 2020 (f).**



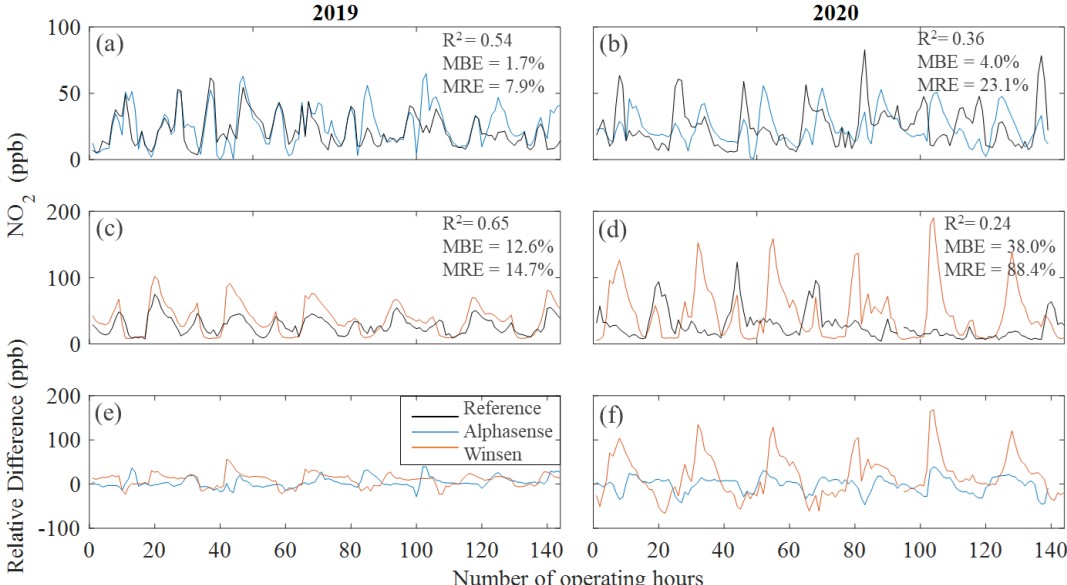

**Figure 10: Time series of hourly-averaged concentrations measured by the Alphasense (a-b) and Winsen (c-d) NO₂ LCSs over six-consecutive days at the beginning (left column; October 2019 for Alphasense and November 2019 for Winsen) and at the end (right column; October 2020 for Alphasense and September 2020 for Winsen) of our measuring period. Last row shows the concentration actual difference between CO LCSs and reference instrument in 2019 (e) and 2020 (f).**

**Table 4: Median values of CO and NO₂ LCSs MREs over 6 consecutive days at the beginning (2019) and at the end (2020) of the measuring period and the respective median values of temperature, RH and reference concentrations.**

| | NMRE | Median Temperature (°C) | | Median RH (%) | | Median Reference Concentration (ppb) | |
|---|---|---|---|---|---|---|---|
| | | 2019 | 2020 | 2019 | 2020 | 2019 | 2020 |
| **Alphasense CO** | -2.6 | 22.3 | 21.6 | 56.2 | 66.8 | 298.0 | 576.7 |
| **Alphasense NO₂** | 1.9 | 21.0 | 20.5 | 64 | 70.1 | 18.4 | 22.4 |
| **Winsen CO** | 2.0 | 12.8 | 26.3 | 74.8 | 61.4 | 458.9 | 649.5 |
| **Winsen NO₂** | 1.3 | 12.8 | 26.3 | 75.0 | 61.65 | 25.6 | 22.2 |




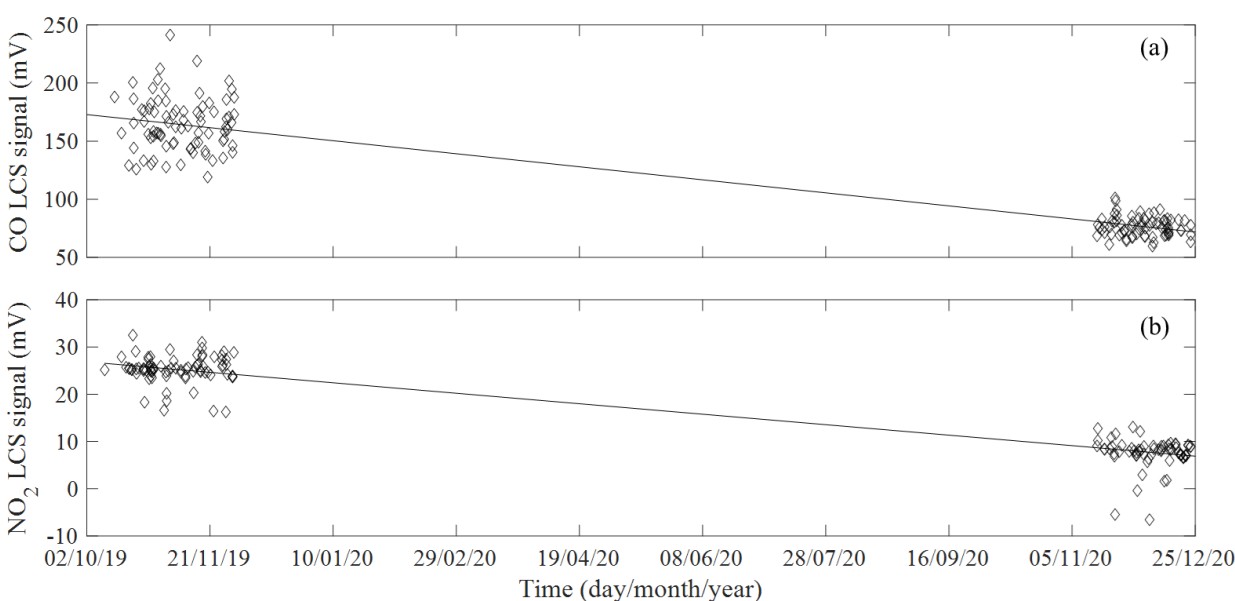

**Figure 11: Signal drift of the (a) CO and (b) NO₂ Alphasense LCSs over a period of a year. The solid line shows the linear signal fitting.**

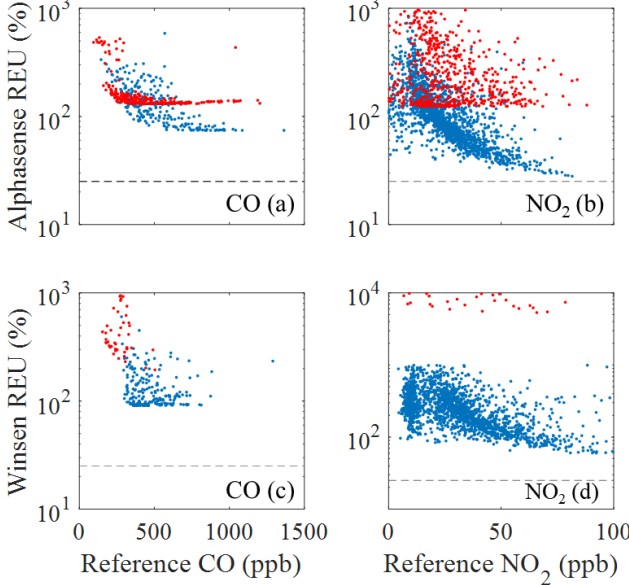

615

**Figure 12: Relative expanded uncertainty of the 8-hour averaged CO (a and c) and the hourly averaged NO₂ (b and d) measurements of the LCSs tested in this study, calculated using the methodology described on the AQ Directive 2008/50/EC. The REU limit is indicated with horizontal dashed black line. Data shown are divided into two time periods: before (blue dots) and after (red dots) summer 2020.**