# Peer review of "Field Evaluation of Low-cost Electrochemical Air Quality Gas Sensors at Extreme Temperature and Relative Humidity Conditions"

_EGUsphere, 2022_

## Author Comment (AC1)

**Anonymous Referee #1, 09 Mar 2023**

**AC:** We thank Referee #1 for her/is time and effort to read in detail our manuscript and provide useful comments. Below are our responses to all the points s/he raised. Necessary modifications in the revised manuscript are noted with red font colour.

**RC1**: What is the reasoning for using two different sensor manufacturers rather than evaluating duplicates from a single manufacturer?

**AC:** This point is well taken. Investigations using multiple low-cost sensors from the same manufacturers have already been reported in the literature (e.g. Spinelle et al., 2015, Zuidema et al., 2021, Castell et al., 2017). Given that, we decided to focus our work on evaluating their performance when exposed to highly variable environmental conditions, with extremely high temperatures (up to 45 °C) and low relative humidity conditions (below 10%) during the summer period, with the goal to make a comparative study between two manufacturers. We believe that our findings, despite that only one sensor type from each manufacturer was used, reveal the issues encountered with those low-cost sensors when used at extreme temperature and relative humidity conditions without requiring using more than one from each manufacturer.

**RC1**: Although the cross-sensitivities are stated in Table S1 and line 258 states that Alphasense provides a correction for this, it is unclear in Eq. 1 and lines 157-159 which variables correspond to the NO2 and the O3 LCSs.

**AC:** We thank the reviewer for pointing this out. We have now updated the text so that the correction for the cross-sensitivity of the O3 LCS is clear. Specifically, we have added the following lines:

"*To account for cross-sensitivity of the $O_3$ LCS from $NO_2$, the first term in the nominator of Eq. 1 becomes $V_{WEu} - V_{WEe} - V_{WEc(NO2)}$, where $V_{WEc(NO2)}$ is the voltage induced on the working electrode, calculated using the $NO_2$ concentration measured from the $NO_2$ LCS, multiplied by the sensitivity (expressed in mV/ppb) of the $O_3$ sensor on $NO_2$, following the guidelines by the manufacturer (Alphasense Application Note AAN 803-05, 2019).*" (cf. Lines 161-164 in the revised manuscript)

**RC1**: Although the Winsen specification sheet does not mention cross-sensitivity I imagine they would have the same cross-sensitivity issues as any other EC sensor. Please discuss why further $NO_2$ and $O_3$ corrections attempts were not made using the available reference instrument data.

**AC:** Indeed, similarly to Alphasense, Winsen LCSs can have cross-sensitivities with other gaseous pollutants (e.g., the significance of $NO_2$ cross-sensitivity to $O_3$ was half the one of the target gas and vice versa). The manufacturer, however, does not provide equations that account for cross-sensitivity, as in the case of Alphasense. Given that the main focus of our work is to evaluate the performance of the sensors as provided by the manufacturer, we did not carry out any further corrections as this is beyond the scope of our study. To clarify that, we have added the following sentence in the updated version of the manuscript:

"*It is well possible that the Winsen $NO_2$ and $O_3$ LCSs are also prone to cross-sensitivities, despite the fact that the manufacturer does not provide any information or a correction for that. Investigating that, however, is beyond the scope of this study, and thus the respective measurements were used as provided by the sensors.*" (cf. Lines 276-278 in the revised manuscript)

**RC1**: Line 177: Please include more information on the chosen temperature and RH regimes and why they were chosen.

**AC:** We very much appreciate the point raised here on the clarity. We have updated line 177 in the original manuscript from:

"*The effect of the environmental parameters (i.e. temperature and RH) on the performance of the sensors was assessed by dividing the whole dataset into different temperature and RH regimes.*"

to:

"*The effect of the environmental parameters (i.e. temperature and RH) on the performance of the sensors was assessed by dividing the whole dataset into different temperature (i.e., < 10, 10-20, 20-30, and > 30 °C) and RH (i.e., < 30, 30-55, 55-75, and > 75%) ranges. These ranges were selected considering the minimum and maximum temperature and RH values encountered during our measurement period and the conditions that the LCSs are calibrated in the laboratory by the manufacturer (i.e., T = 20 °C and RH = 60%).*" (cf. Lines 188-192 in the revised manuscript)

**RC1**: What is the scientific reasoning for subtracting 400ppb from the Winsen CO sensors? Is this just a manufacturer issue? Line 213 explaining this offset could be moved to Section 2.3 line 173.

**AC:** This is a consistent drift observed from the beginning of the measuring period, and most probably is related to manufacturer issues (e.g., calibration, firmware, proprietary algorithms, etc.). The sentence explaining this offset has now been moved to Section 2.3. The following lines were updated from:

"*Instead, measurements from a built-in temperature sensor are used internally to capture respective variabilities, providing signal outputs that are directly expressed as concentrations (ppb).*"

to

"*Instead, measurements from a built-in temperature sensor are used internally to capture respective variabilities, providing signal outputs that are directly expressed as concentrations (ppb). It should be noted that an offset of ca. 400 ppb in the raw measurements from the Winsen CO LCS was observed throughout the entire period of our study, and that this value has been subtracted from the reported data to facilitate comparison with the measurements by the respective reference instrument.*" (cf. Lines 179-183 in the revised manuscript)

**RC1**: Line 227: It is mentioned that LCSs overestimated NO2 concentrations, which could be due to their ozone cross-sensitivity as this was not corrected for, this point is only brought up later in line 279. Perhaps start this discussion earlier in line 227.

**AC:** We agree with the suggested correction here. This point is now brought up earlier (line 240 in the revised version of the manuscript, which was line 227 in the old version of manuscript) as suggested.

**RC1**: Table 1: What do you mean by "useful measurements"?

**AC:** We agree that the word "*useful*" can be misleading. We clarify that those were "*the number of hourly averaged data points used for data analysis of each LCS*" in the updated version of the manuscript. This correction was adopted also in Tables 2 and 3 in the revised manuscript.

**RC1**: Line 221: While the mean CO values between the LCSs and references instruments are within error the Winsen CO has low R2 and both have low slopes in Figure 4a & 4e and Table 2 so I'm not

sure I agree that there is "good agreement" between these LCSs and the reference instrumentation, maybe relative to the other LCSs tested. The U.S. EPA Air Sensor Performance Targets and Testing Protocols establish statistical targets for LCSs (currently Ozone and PM have been published) which includes a bias performance metric with slope and intercept calculations. Performance targets for NO2, CO and SO2 are being deliberated. It would be useful to include these metrics in Table 1.

**AC:** We do see the point raised here by the reviewer. The phrase "*good agreement*" is now corrected to "*better agreement compared to the rest of sensors tested*" (cf. lines 231-232 in the revised manuscript).

In addition, the slope and intercept are now included in Table 1 of the revised manuscript as per the U.S. EPA Air Sensor Performance Targets and Protocols (Duvall et al., 2021).

**RC1**: Line 247: Please state the temperature and RH conditions for winter and summer periods respectively and how you define summer and winter (date ranges).

**AC:** The temperature and RH conditions for the winter and summer period are now added (line 258-261 in the revised manuscript). The following lines were updated from:

"*We should note here that the temperatures and RH conditions that the LCSs were exposed to during the entire measuring campaign ranged respectively from 3 to 43 °C and from < 10 to 97%.*"

to:

"*We should note here that the temperatures and RH conditions that the LCSs were exposed to during the entire measuring campaign ranged respectively from -0.4 to 42.4 °C and from 2.8 to 96.8%. More specifically, the temperature ranged from 14.1 to 42.4 °C during summer (Jun.-Aug. 2020) and from -0.4 to 21.3 °C during winter (Dec. 2019-Feb. 2020), while the RH varied from 2.8 to 95.2%, and from 32.7 to 96.8%, during summer and winter, respectively.*"

**RC1**: Line 377: Please correct to "making it the only sensor that can"

**AC:** The suggested change is unclear. The sentence in the manuscript reads:

"*The same is true for the Winsen, but not for the Alphasense $NO_2$ sensor which appears to reach this threshold at concentrations higher than ca. 70 ppb (cf. Fig. 12b), making it the only sensor that can qualify for indicative measurements according to the EC directive.*"

We believe this sentence is clear. No further action is taken here.

**RC1**: SI Line 37: Please correct to "the further we move below"

**AC:** The correction has been made. The phrase "*the furthest we move below*" has now been changed to "*the further we move below*".

**RC1**: Table S1: In the top row an error pops up in the PDF version "[Error! Reference source not found.]" Was a picture supposed to placed here?

**AC:** This was literature citation software related issue. The error has been corrected.

**RC1**: Please include more specific information on the Peltier et al. 2020 reference.

**AC:** We see the point of the reviewer. The reference has been updated.

References:

Castell, N., Dauge, F.R., Schneider, P., Vogt, M., Lerner, U., Fishbain, B., Broday, D., Bartonova, A.: Can commercial low-cost sensor platforms contribute to air quality monitoring and exposure estimates?, Environ. Int., 99, 293-302, 10.1016/j.envint.2016.12.007, 2017.

Duvall, R.M., Hagler, G.S.W., Clements, A.L., Benedict , K., Barkjohn, K., Kilaru, V., Hanley, T., Watkins, N., Kaufman, A., Kamal, A., Reece, S., Fransioli, P., Gerboles, M., Gillerman, G., Habre, R., Hannigan, M., Ning, Z., Papapostolou, V., Pope, R., Quintana, P.J.E., Lam Snyder, J.: Deliberating Performance Targets: Follow-on workshop discussing PM10, NO2, CO, and SO2 air sensor targets, Atmos. Environ., 246, 118099, 10.1016/j.atmosenv.2020.118099, 2021.

Spinelle, L., Gerboles, M., Villani, M.G., Aleixandre, M., Bonavitacola, F.: Field calibration of a cluster of low-cost available sensors for air quality monitoring. Part A: ozone and nitrogen dioxide, J. Sens. Actuators B Chem., 215, 249-257, doi:10.1016/j.snb.2015.03.031, 2015.

Zuidema, C.; Schumacher, C.S.; Austin, E.; Carvlin, G.; Larson, T.V.; Spalt, E.W.; Zusman, M.; Gassett, A.J.; Seto, E.; Kaufman, J.D.; Sheppard, L.: Deployment, Calibration, and Cross-Validation of Low-Cost Electrochemical Sensors for Carbon Monoxide, Nitrogen Oxides, and Ozone for an Epidemiological Study, Sensors, 21, 4214, doi.org/10.3390/s21124214, 2021.

---

## Author Comment (AC2)

**Anonymous Referee #2, 13 Mar 2023**

**AC:** We thank Referee #2 for her/is time and effort to go over our manuscript and provide useful comments. Below are our responses to all the points s/he raised. The associated modifications in the revised manuscript are noted with red font colour.

**RC2**: You mention that you use the reference thermometers at the station to compute the different correction factors. Do you think that there can be a difference between the temperature recorded by the reference thermometers and the real temperature at which the sensor box is exposed? Can it impact the correction factors calculations or the final concentrations? Are you including temperature sensors in the sensor box?

**AC:** A temperature/RH sensor was included in the two AQ monitors, and there was a big difference between the measurements of these sensors compared to the respective reference ones, especially during high temperature days. The reason of using the ambient temperature (rather than the temperatures recorded by the sensors inside the air quality monitor) was that the sensing area of the sensors was exposed to the ambient environment. In fact, we explored that using the temperatures recorded by the sensors located inside the AQ monitors for converting the LCS signals to concentrations, but that resulted to much higher deviations from the reference measurements.

To address and clarify this point, we have updated the following lines from:

"*The temperatures used as input for the calculation of the correction factors $n_T$ and $k_T$ are the ones measured by the reference thermometers at the station.*"

to

"*While a temperature/RH sensor was included inside the two AQ monitors, the temperatures used as input for the calculation of the correction factors $n_T$ and $k_T$ are the ones measured by the reference thermometers at the station. We should note here that the temperature and RH sensor used in the two AQ monitors, recorded measurements that deviated substantially from those reported by the reference sensors, and thus were not used for determining the concentrations from the LCSs.*" (lines 173-176 in the updated version of the manuscript).

**RC2**: Table S1 shows some sort of compilation errors "Error! Reference source not found".

**AC:** We thank the reviewer for pointing this out. The error has now been corrected. This was literature citation software issue.

**RC2**: Many recent works use a machine learning-based in-situ calibration, comparing the LCS measurements with the reference instrument. Do you think that some sort of sensor recalibration where the sensor calibration model is trained with data with large temperatures may reduce the worsening of the sensor?

**AC:** Indeed, this is the case. In this work we focused only on the performance evaluation of low-cost electrochemical gas sensors at extreme temperature and RH conditions, using the calibration models provided by the manufacturer. In fact, we are currently exploring the effectiveness of machine learning algorithms for calibrating measurements from LCSs, including identifying the best calibration schemes one can follow in doing so. We deliberately did not include any results from that work in this manuscript, as it would deviate from its main focus. We plan to submit a follow-up paper on the results from the work we did on the use of machine learning algorithms very soon.

**RC2**: As noticed in Table I, different sensors have different numbers of useful samples (N), the difference can be up to 2000. Is it due to missing data, sensor issues?

**AC:** This point is well taken. The difference in the number of hourly averaged data points used for deriving the performance of each LCS is attributed to the following two reasons: (a) missing data points due to data acquisition systems malfunctions and/or overheating of the sensors electronics, and (b) omission of negative concentrations (i.e., measured by the LCSs). We should note that in the case of the Winsen LCSs, less measurements were recorded during summer in comparison to the Alphasense LCSs, because their operation was interrupted earlier than the Alphasense due to electronics overheating.

We believe that these points are clear in the manuscript. For instance, in Lines 304-306 that read:

"*It should be noted that even though the sensors were not operating during the summer months, they were exposed to ambient temperatures (up to 45 °C), while their main body containing the electrolyte was at even higher values (i.e., > 50 °C) during midday, as a result of heat built up within the cases of the AQ monitors.*"

and in lines 352-355:

"*This was feasible for the datasets of the Alphasense sensors but not so easy for the Winsen, as those were not always functional after the summer period, yielding significantly less measurements during that period as indicated in Table 1 and reflected by the high temperature differences in the two periods analysed here (cf. Table 4).*"

**RC2**: In Figure 3, it can be seen how the O3 estimation for the Alphasense sensor completely overestimates the reference values. Does it make sense? As far as I understand this period corresponds to beginning of the testing and the temperature is not large is this case.

**AC:** Even though LCSs are proven to be highly influenced by high temperature and low RH conditions, the performance of Alphasense $O_3$ sensor is poor from the beginning of the measuring period. This is possibly due to the cross-sensitivity of $O_3$ sensor to $NO_2$, as mentioned in section 3.1 (lines 272-277 in the original manuscript; now lines 270 – 275 in the revised manuscript) that read:

"*The high errors of the Alphasense $O_3$ measurement can be associated to interferences with $NO_2$. This is in fact taken into account, as the signal from the $NO_2$ LCS (mV) is subtracted from the $O_3$ sensor signal following the guidelines by the manufacturer (cf. first term in the nominator of Eq. (1)). Although this correction is used for the performance analysis of the Alphasense $O_3$ LCS, it yields a measurement error that is higher compared to the case when the correction is not applied (cf. Fig. S6), most likely due to error propagation from the signal of the $NO_2$ sensors to the reported concentrations of $O_3$*"

We believe that point is addressed, and thus no further action is taken.

**RC2**: Have you evaluated/shown the possible joint effect of temperature and relative humidity? It would be interesting to show a result by diving the data into, low T+low RH, low T + high RH, etc.

**AC:** As stated in the manuscript, during the period of our study the low temperatures are typically accompanied with high RH values and vice versa (cf. Fig. S5 in the supplement and Figure 1 below). As shown by measurements (Figure 1 below), there are not many cases (data points) with low temperature and low RH (bottom left corner in the plot), or high temperature and high RH (top right corner) to proceed with the comparative analysis suggested by the reviewer. Analysis of the other

cases is already provided in the manuscript, so no further action was taken with respect to this point.

[Figure]

**Figure 1:** Correlation between the temperature and RH measurements recorded by the reference monitoring station for the whole measurement period.